

# Comparative analysis of lingual bracket transfer accuracy using fully versus partially enclosed 3D-printed indirect bonding trays: an *in vivo* study

Viet Anh Nguyen[1,2]

[1] Faculty of Dentistry, Phenikaa University, Hanoi, Vietnam
[2] Viet Anh Orthodontic Clinic, Hanoi, Vietnam

## ABSTRACT

**Background:** Lingual orthodontic treatment requires precise bracket positioning to ensure optimal outcomes. This study aimed to compare the transfer accuracy of fully enclosed (FE) and partially enclosed (PE) three-dimensionally (3D) printed indirect bonding trays for lingual brackets, focusing on linear and angular errors, and to evaluate their clinical applicability.

**Materials and Methods:** A total of 20 patients with 498 brackets bonded on both arches were included in this *in vivo* study. Two tray designs—FE and PE—were fabricated using a standardized digital workflow and 3D-printed with rigid resin. Bracket transfer accuracy was assessed by superimposing post-bonding scans with planned virtual models. Linear (mesiodistal, buccolingual, occlusogingival) and angular (rotation, angulation, torque) errors were measured. Statistical analyses included t-tests to compare transfer accuracy between the two tray designs.

**Results:** Both tray designs achieved clinically acceptable linear accuracy, with errors below 0.5 mm across all dimensions. Angular errors for rotation and angulation were also within clinically acceptable limits of 2°, but torque transfer remains a challenge for both tray designs. The PE design showed significantly lower buccolingual and occlusogingival errors for premolars and the total sample, while the FE design demonstrated significantly lower torque errors for molars. However, the FE design required longer bonding times (84.2 ± 14.5 min) compared to the PE design (70.7 ± 12.8 min, $p = 0.041$), without offering a significant overall accuracy advantage.

**Conclusion:** The PE tray design is the preferred option over the FE design for its simplicity and accuracy. Torque errors in the PE design can be mitigated with supplementary stabilization materials.

# INTRODUCTION

Lingual orthodontic treatment has gained increasing popularity in recent years due to its esthetic advantages and the development of customized appliances. Unlike labial brackets, the unique morphology of lingual surfaces and limited direct access to these areas make accurate bracket positioning crucial for achieving optimal treatment outcomes. Indirect

Corresponding author
Viet Anh Nguyen, anh.
nguyenviet1@phenikaa-uni.edu.vn

bonding has become the standard method for transferring predetermined bracket positions from setup models to the patient's dentition, providing greater precision in alignment and reducing operator dependency during bonding (*Nguyen, 2024*; *Schubert et al., 2013*).

Advances in digital technology have revolutionized orthodontic workflows, enabling virtual setups and the use of three-dimensionally (3D)-printed transfer trays. This digital approach eliminates many challenges associated with traditional analog methods. Studies have shown that 3D-printed indirect bonding trays for labial brackets achieve high accuracy in transferring linear dimensions, such as mesiodistal, buccolingual, and occlusogingival positions, while angular dimensions, including torque, angulation, and rotation, remain less consistently transferred (*Niu et al., 2021*; *Xue et al., 2020*; *Hoffmann et al., 2022*; *Bachour, Klabunde & Grünheid, 2022*). Despite these advancements, limited attention has been given to optimizing tray designs for lingual brackets.

Specifically, the design of the 3D-printed tray—fully enclosed (FE) *vs* partially enclosed (PE)—plays a critical role in clinical performance. Fully enclosed designs offer better bracket retention and stability but may hinder seating during bonding and complicate adhesive removal (*Nguyen, 2024*). Partially enclosed designs, while allowing better visualization, may compromise bracket stability during transfer (*Schwärzler et al., 2023*; *Kim, Chun & Kim, 2018*). These trade-offs highlight the need for a systematic evaluation of these tray designs, particularly for lingual orthodontics where the transfer process is more complex due to the unique challenges posed by lingual surfaces.

Previous studies have investigated the precision of 3D-printed indirect bonding trays for labial brackets, but research focusing on their application in lingual orthodontics remains sparse. *Schubert et al. (2013)* evaluated traditionally fabricated analog jigs for lingual brackets, reporting linear transfer accuracy ranging from 0.10 to 0.13 mm and angular discrepancies between 2.20° and 3.31°. Similarly, *Anh et al. (2024)* assessed vacuum-formed trays fabricated on 3D-printed models, demonstrating linear transfer accuracy from 0.06 to 0.12 mm and angular errors from 1.28° to 2.96°. Moreover, existing research on tray designs has largely overlooked the potential impact of enclosure variations on the accuracy of bracket transfer for lingual orthodontics.

Early 3D-printed indirect bonding trays for labial brackets were first fabricated from flexible photopolymer resins, facilitating tray removal but permitting micro-movements that reduced transfer accuracy (*Niu et al., 2021*; *Bachour, Klabunde & Grünheid, 2022*; *Palone et al., 2023*). Rigid printable resins were later introduced to enhance stability, although their reduced flexibility could hinder tray seating, particularly in crowded arches (*Niu et al., 2021*). Tray enclosure design also affects accuracy, with fully enclosed trays typically yield smaller torque errors but may complicate seating, while partially enclosed designs facilitate bonding but risk bracket instability (*Bachour, Klabunde & Grünheid, 2022*). Although previous studies have discussed modifications from full to partial enclosure to address tray cracking during bracket embedding, difficulties in tray removal, and increased bond failure rates, a systematic comparison of these enclosure strategies has not yet been conducted, especially for lingual brackets. In addition, variables such as tray thickness, offset, and build angle, which influence tray rigidity and polymerization

shrinkage, have been studied mainly *in vitro* (*Hoffmann et al., 2022*; *Kim, Chun & Kim, 2018*; *Koch et al., 2022*; *Eglenen & Karabiber, 2024*).

Given the paucity of clinical data on lingual indirect bonding tray designs, this study aims to compare the transfer accuracy of fully *vs* partially enclosed designs of 3D-printed trays *in vivo*, while controlling for segmentation span and offset. The null hypothesis is that there is no significant difference in the transfer accuracy of these two tray designs in terms of both linear and angular bracket positioning errors. By exploring these designs, the study seeks to optimize bonding precision and guide clinicians in selecting the most effective approach for lingual orthodontic treatment.

## MATERIALS AND METHODS

### Subject

This study was conducted in compliance with ethical standards and approved by the Hanoi Medical University Institutional Ethical Board Review (approval no. HMUIRB970). The research was designed to adhere to the STROBE guidelines and comply with all applicable regulations throughout its implementation. All study procedures were carried out in compliance with applicable guidelines and regulations.

Participants were recruited consecutively from a private orthodontic practice specializing in lingual appliances (Nam Tu Liem, Hanoi) between September 12th 2023 and February 12th 2024. The inclusion criteria were as follows: (1) individuals aged 18 years or older; (2) presence of a complete permanent dentition (excluding third molars); (3) no dental anomalies, such as hypodontia, supernumerary teeth, or tooth malformations; and (4) no history of previous orthodontic treatment. The exclusion criteria included: (1) teeth with restorations or prosthetic crowns; (2) active periodontal disease; (3) craniofacial syndromes or systemic diseases affecting dental structures; and (4) inability to provide informed consent. All patients meeting the eligibility criteria during the study period were enrolled without exclusions. Patients received an information sheet explaining the study objectives, procedures, and potential risks, and written informed consent was obtained from all participants prior to enrollment.

The sample size was determined based on a power calculation. Using the effect size of 0.604 derived from torque transfer errors reported in the study by *Wang et al. (2023)*, where two types of 3D-printed trays exhibited torque transfer errors of $0.83° \pm 0.46°$ and $1.12° \pm 0.50°$, the minimum required sample size was calculated at a significance level of 0.05 and 95% power. The calculation indicated that a minimum of 73 brackets per group needed to be analyzed. Torque was chosen for the sample size calculation because prior studies on labial (*Niu et al., 2021*; *Bachour, Klabunde & Grünheid, 2022*) and lingual (*Anh et al., 2024*) brackets have shown that errors in torque are typically the highest among angular discrepancies.

### Tray fabrication and appliance bonding

The design and fabrication of fully and partially enclosed 3D-printed indirect bonding trays adhered to a systematic digital workflow (*Nguyen, 2024*). Virtual models with segmented teeth of the patient's dentition were created from intraoral scans (i700; Medit,

Seoul, Korea) using the Autolign orthodontic software (Diorco, Gyeonggi-do, Korea). Within Autolign, an ideal setup model was generated and brackets (ADB; Medico, Gyeonggi-do, Korea) were virtually positioned on the aligned dentition with guidance from a virtual archwire, ensuring optimal alignment and positioning relative to the lingual tooth surfaces (clearance ≥ 0.03 mm). These bracket positions were then transferred back to the initial malocclusion model to facilitate the design of indirect bonding trays. The selected brackets are already included in the Autolign library, and the manufacturer additionally provided the original 3D design files, allowing accurate matching and deviation measurements in later steps.

For tray design, the brackets were extruded toward the teeth to block out the space between the bracket bases and the lingual tooth surfaces, as well as undercuts below the bracket wings. Patients were consecutively allocated into two groups: the FE design, which encapsulated the brackets completely, and the PE design, which featured trimmed gingival walls to expose the bottom of the gingival bracket wings (Fig. 1). Both tray designs were created with a uniform thickness of 1.5 mm and segmented into two- to three-tooth spans to minimize transfer errors caused by tray bending due to polymerization shrinkage of the rigid tray material. For the six anterior teeth, the trays were typically divided into two segments of three teeth each, except in cases where severe crowding necessitated omitting a tooth from bonding. For posterior teeth, segmentation depended on the extraction pattern: in extraction cases, posterior trays included three-tooth spans; in nonextraction cases, the trays were divided into two segments, each containing two teeth. Additionally, the offset between the tray and the tooth-bracket combination was set at 0.03 mm to ensure precise adaptation while compensating for errors due to resin shrinkage without hindering proper seating (*Nguyen, 2024*). Tray extrusion, generation, and trimming were performed using Medit Design (Medit, Seoul, Korea) and Meshmixer software (Autodesk, San Francisco, CA, USA).

The completed tray designs were sliced and prepared for 3D printing using a Photon D2 digital light processing (DLP) 3D printer (Anycubic, Shenzhen, China) with Surgical Guide rigid resin (Ludent, Gyeonggi-do, Korea), which has a Shore D hardness of 67. Fully enclosed trays were printed at a 110° build angle to provide adequate support for the gingival walls, while partially enclosed trays were printed at a 180° build angle to reduce printing time. All trays were printed using a layer thickness of 0.03 mm and a layer exposure time of 2.5 s, following the resin manufacturer's recommended parameters. After printing, the trays were cleaned with 99% isopropyl alcohol for 6 min and then post-cured for 5 min under 405-nm ultraviolet light using the Wash & Cure 3 system (Anycubic, Shenzhen, China).

During the clinical bonding procedure, the patient's dentition was cleaned with non-fluoridated pumice (Prophy Paste; Ortho Technology, San Diego, CA, USA), etched with FineEtch etchant (Spident, Gyeonggi-do, Korea), and primed with Assure Plus primer (Reliance, Chicago, IL, USA). GoTo adhesive (Reliance, Chicago, IL, USA) was applied to the bracket bases, and the trays were seated with firm pressure to ensure proper placement. Light curing using a LedF curing light (Woodpecker, Guilin, China) secured the brackets in place, after which the trays were carefully removed. After curing and securing the

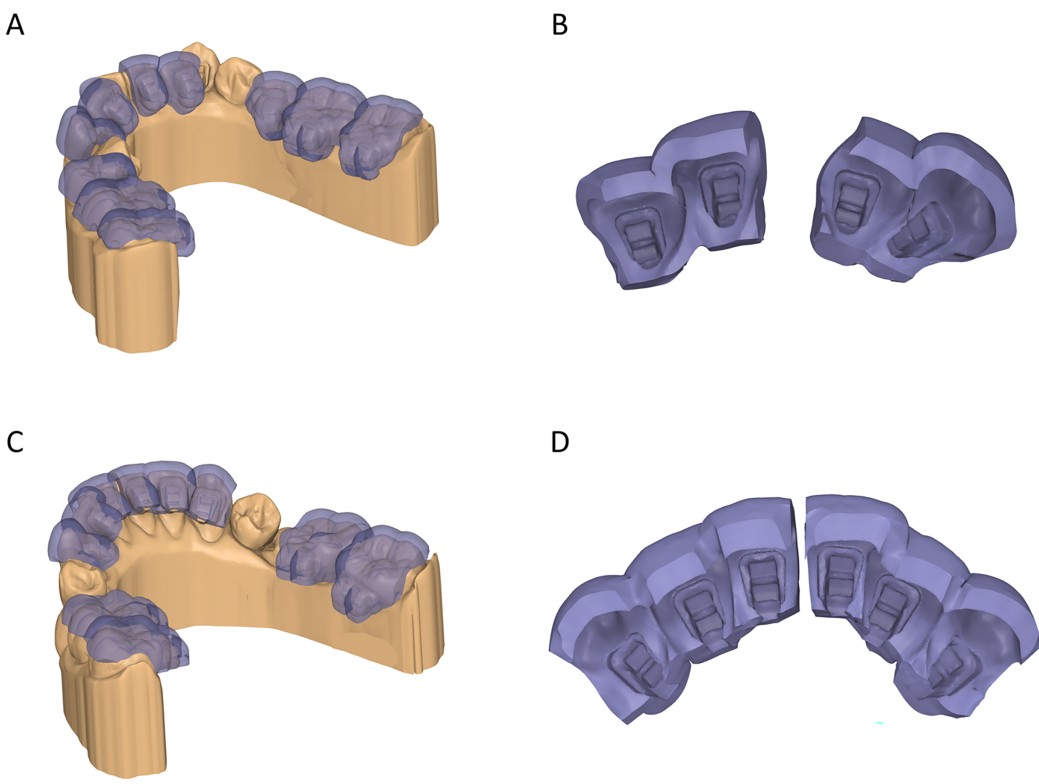

**Figure 1 Designs of 3D-printed indirect bonding trays.** (A, B) Fully enclosed design. (C, D) Partially enclosed design.

brackets in place, additional trimming of the tray material surrounding the brackets was performed as needed for both tray types to facilitate easy removal, addressing the rigidity of the resin used. After tray removal, excessive adhesives were removed using a pointed bur (CD-53F; Mani, Tochigi, Japan), finishing discs (Sof-Lex, 3M, Seefeld, Germany), and silicone polishers (One Gloss; Shofu, Kyoto, Japan). Additionally, bonding time was recorded as the duration from the placement of the first tray segment onto the teeth to the completion of excess adhesive removal.

## Transfer error measurement

Bracket transfer error measurement was performed by scanning the patient's bonded dental arches using the same intraoral scanner (i700) and superimposing them with the virtual models containing the planned bracket positions. The discrepancies in bracket positions were measured using Meshmixer software (Autodesk, San Francisco, CA, USA) following the approach of *Koch et al. (2022)*. To ensure precise matching, the data segments were processed by isolating each tooth and its corresponding bracket, excluding surrounding gingival structures. For each bracket, the planned bracket position was set at the origin of a coordinate system, with the x-axis parallel to the mesiodistal edge of the bracket slot, the z-axis parallel to the buccolingual edge of the bracket slot, and the y-axis parallel to the vertical axis of the bracket (Fig. 2).

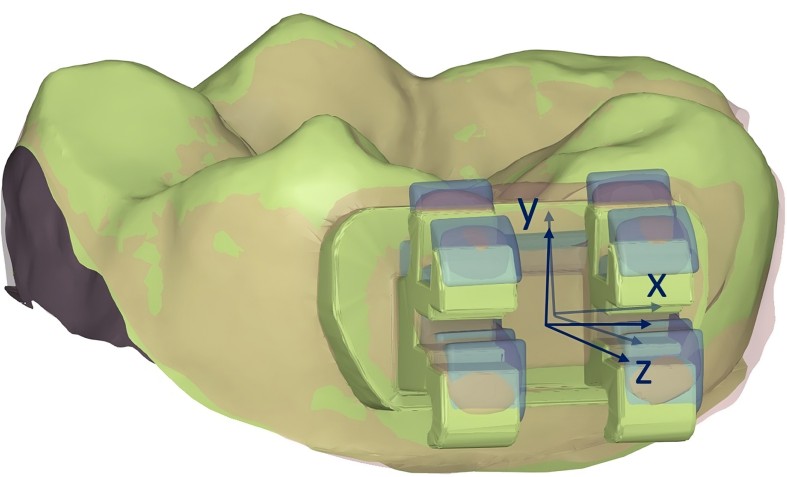

**Figure 2 Measurement of bracket transfer errors.**

Virtual bracket patches were first aligned with the scanned brackets on the post-bonding data. Subsequently, both bracket patches and post-bonding data were aligned to the corresponding tooth surface from the reference data, excluding the bracket itself. The coordinate differences of the virtual bracket patches relative to the planned setup were then calculated, representing the bracket transfer deviations. Three linear dimensions were measured including mesiodistal, buccolingual, and occlusogingival measurements. Three angular dimensions were also measured including rotation, angulation, and torque. Rotation referred to the rotation of the bracket in the xz plane, angulation referred to the rotation in the xy plane, and torque referred to the rotation in the yz plane.

## Statistical analysis

Absolute values of the discrepancies were calculated to avoid positive and negative values negating each other. The Smirnov-Kolmogorov test was used to assess the normality of data distribution. Inter- and intra-observer reliability was evaluated using both the intraclass correlation coefficient (ICC) and Bland-Altman analysis. For inter-observer reliability, two independent investigators independently repeated the entire matching and measurement process on a subset of 50 brackets that were randomly selected from the total sample. For intra-observer reliability, one investigator re-performed the entire matching and measurement process on the same subset after a two-week interval. ICCs were calculated based on a two-way random-effects model, absolute agreement, and single measurement definition. Bland-Altman analysis was also conducted to assess systematic and random measurement errors for each linear and angular parameter.

Brackets were further divided into tooth types, including anterior, premolar, and molar brackets. Descriptive statistics, including means and standard deviations, were calculated for each bracket type and the whole sample. One-sample t-tests were used to determine whether the mean bracket deviations were significantly lower than the clinically acceptable limits of 0.5 mm for linear dimensions and 2° for angular dimensions for each tray type (*Niu et al., 2021*; *Hoffmann et al., 2022*; *Bachour, Klabunde & Grünheid, 2022*;
*Anh et al., 2024*). Two-sample t-tests were employed to detect significant differences in bracket transfer errors between the two tray types. The significance level for all statistical tests was set at $\alpha = 0.05$.

## RESULTS

A total of 20 patients were included in the study, with 498 brackets bonded on both the upper and lower arches. Demographic characteristics of the study groups are presented in Table 1. The mean age of participants was $30.4 \pm 6.6$ years in the fully enclosed tray group and $28.5 \pm 7.4$ years in the partially enclosed tray group. The mean space deficiency was $3.81 \pm 3.97$ and $4.03 \pm 2.61$ mm in the fully and partially enclosed tray groups, respectively. No statistical differences were found in the mean age ($p = 0.554$), gender ($p = 0.639$), extraction or non-extraction treatment ($p = 0.653$), or amount of crowding ($p = 0.768$) between the two groups. In contrast, the total bonding time was significantly longer ($p = 0.041$) in the FE group ($84.20 \pm 14.54$ min) compared to the PE group ($70.70 \pm 12.79$ min). Three brackets in the FE group were excluded due to bond failure, resulting in 242 brackets analyzed in this group, while all 256 brackets were included in the PE group. However, there was no statistical difference in the failure rate between the two tray designs ($p = 0.236$).

The inter-observer ICC values ranged from 0.906 to 0.944 for linear measurements and from 0.905 to 0.918 for angular measurements. The intra-observer ICC values ranged from 0.924 to 0.949 for linear measurements and from 0.914 to 0.927 for angular measurements, indicating excellent reliability for both inter- and intra-observer assessments. Bland-Altman analyses showed minimal mean biases, ranging from 0.001 to 0.011 mm for linear measurements and from 0.159° to 0.375° for angular measurements. The 95% limits of agreement for linear measurements were within ±0.1 mm, and for angular measurements were within approximately ±3° (Table 2).

Bracket transfer errors for each tooth type and the total sample for the fully enclosed and partially enclosed trays are presented in Tables 3 and 4, respectively. For linear dimensions, both groups achieved mean errors significantly less than the clinically acceptable limit of 0.5 mm across all tooth types and the total sample ($p < 0.001$). Mesiodistal errors ranged from $0.07 \pm 0.06$ to $0.10 \pm 0.10$ mm, buccolingual errors from $0.07 \pm 0.06$ to $0.12 \pm 0.10$ mm, and occlusogingival errors from $0.10 \pm 0.08$ to $0.19 \pm 0.15$ mm (Fig. 3). For angular dimensions, rotation and angulation errors were significantly less than the clinically acceptable limit of 2° across all tooth types and the total sample for both groups ($p < 0.01$) except for premolars in the FE group ($p > 0.05$). Torque errors were not statistically within the clinically acceptable limit for both groups ($p > 0.05$). Rotation errors ranged from $1.02° \pm 0.97°$ to $1.88° \pm 1.66°$, angulation errors from $1.12° \pm 1.03°$ to $1.70° \pm 1.61°$, and torque errors from $2.10° \pm 1.88°$ to $3.39° \pm 2.69°$ (Fig. 4).

Comparisons of bracket transfer errors between the two tray designs are presented in Table 5. No significant differences were found in mesiodistal, rotation, and angulation errors between the two groups for any tooth type or the total sample ($p > 0.05$). Significant differences were observed in buccolingual errors for premolars ($p = 0.042$) and the total sample ($p = 0.047$), with the PE group exhibiting lower errors. Similarly, occlusogingival

**Table 1 Demographic characteristics of the study groups.**

| Characteristic | Fully enclosed trays (n = 10) | Partially enclosed trays (n = 10) | p |
|---|---|---|---|
| Age | 30.4 ± 6.6 | 28.5 ± 7.4 | 0.554 |
| Sex | | | 0.639 |
| Male | 3 (30%) | 4 (40%) | |
| Female | 7 (70%) | 6 (60%) | |
| Extraction | | | 0.653 |
| Non-extraction | 4 (40%) | 5 (50%) | |
| Extraction | 6 (60%) | 5 (50%) | |
| Space deficiency | 3.81 ± 3.97 | 4.03 ± 2.61 | 0.841 |
| Upper arch (mm) | 2.80 ± 3.42 | 2.77 ± 2.57 | 0.983 |
| Lower arch (mm) | 4.82 ± 4.39 | 5.28 ± 2.06 | 0.768 |
| Bonding time (min) | 84.20 ± 14.54 | 70.70 ± 12.79 | 0.041* |
| Upper arch (min) | 40.80 ± 7.86 | 33.00 ± 7.53 | 0.036* |
| Lower arch (min) | 43.40 ± 7.35 | 37.70 ± 6.27 | 0.079 |
| Teeth | | | – |
| Included in study | 245 | 256 | |
| Excluded (bond failure) | 3 | 0 | 0.236 |
| Included in analysis | 242 | 256 | |

**Note:**
* Statistically significant difference between tray designs ($p < 0.05$).

**Table 2 Reliability analysis results based on intraclass correlation coefficient and Bland-Altman analyses for transfer error measurements.**

| | n | Intra-observer | | | Inter-observer | | |
|---|---|---|---|---|---|---|---|
| | | ICC | Bias | Limit of agreement | ICC | Bias | Limit of agreement |
| Mesiodistal (mm) | 50 | 0.949 | 0.001 | −0.041 to 0.044 | 0.944 | 0.003 | −0.044 to 0.039 |
| Buccolingual (mm) | 50 | 0.924 | 0.009 | −0.037 to 0.056 | 0.906 | 0.007 | −0.049 to 0.064 |
| Occlusogingival (mm) | 50 | 0.925 | 0.004 | −0.085 to 0.092 | 0.909 | 0.011 | −0.089 to 0.111 |
| Rotation (°) | 50 | 0.927 | 0.212 | −0.587 to 1.011 | 0.918 | 0.182 | −0.669 to 1.034 |
| Tip (°) | 50 | 0.919 | 0.159 | −0.984 to 1.301 | 0.905 | 0.164 | −1.038 to 1.365 |
| Torque (°) | 50 | 0.914 | 0.257 | −2.197 to 2.711 | 0.908 | 0.375 | −2.330 to 3.080 |

**Table 3 Bracket transfer errors of the fully enclosed design for each tooth type and the total sample.**

| Fully enclosed trays | Anterior (n = 109) | Premolar (n = 56) | Molar (n = 77) | Total (n = 242) |
|---|---|---|---|---|
| Mesiodistal (mm) | 0.08 ± 0.07 | 0.08 ± 0.08 | 0.08 ± 0.08 | 0.08 ± 0.07 |
| p | <0.001* | <0.001* | <0.001* | <0.001* |
| Buccolingual (mm) | 0.12 ± 0.10 | 0.10 ± 0.08 | 0.12 ± 0.08 | 0.11 ± 0.09 |
| p | <0.001* | <0.001* | <0.001* | <0.001* |
| Occlusogingival (mm) | 0.11 ± 0.09 | 0.19 ± 0.15 | 0.13 ± 0.11 | 0.14 ± 0.12 |
| p | <0.001* | <0.001* | <0.001* | <0.001* |
| Rotation (°) | 1.41 ± 1.48 | 1.88 ± 1.66 | 1.02 ± 0.97 | 1.40 ± 1.42 |

| Table 3 (continued) | | | | |
| --- | --- | --- | --- | --- |
| Fully enclosed trays | Anterior (*n* = 109) | Premolar (*n* = 56) | Molar (*n* = 77) | Total (*n* = 242) |
| *p* | <0.001* | 0.303 | <0.001* | <0.001* |
| Angulation (°) | 1.68 ± 1.41 | 1.70 ± 1.61 | 1.12 ± 1.30 | 1.50 ± 1.44 |
| *p* | 0.009 | 0.083 | <0.001* | <0.001* |
| Torque (°) | 2.41 ± 2.29 | 3.08 ± 3.10 | 2.16 ± 2.07 | 2.48 ± 2.45 |
| *p* | 0.966 | 0.994 | 0.999 | 0.999 |

Note:
  * Bracket errors were significantly less than 0.5 mm and 2° as indicated by one-sided t-tests.

Table 4 **Bracket transfer errors of the partially enclosed design for each tooth type and the total sample.**

| Partially enclosed trays | Anterior (*n* = 115) | Premolar (*n* = 62) | Molar (*n* = 79) | Total (*n* = 256) |
| --- | --- | --- | --- | --- |
| Mesiodistal (mm) | 0.07 ± 0.06 | 0.10 ± 0.10 | 0.08 ± 0.07 | 0.08 ± 0.07 |
| *p* | <0.001* | <0.001* | <0.001* | <0.001* |
| Buccolingual (mm) | 0.11 ± 0.08 | 0.07 ± 0.06 | 0.10 ± 0.09 | 0.10 ± 0.08 |
| *p* | <0.001* | <0.001* | <0.001* | <0.001* |
| Occlusogingival (mm) | 0.10 ± 0.08 | 0.14 ± 0.11 | 0.13 ± 0.10 | 0.12 ± 0.10 |
| *p* | <0.001* | <0.001* | <0.001* | <0.001* |
| Rotation (°) | 1.47 ± 1.65 | 1.69 ± 1.40 | 1.10 ± 0.84 | 1.41 ± 1.39 |
| *p* | <0.001* | 0.043* | <0.001* | <0.001* |
| Angulation (°) | 1.52 ± 1.13 | 1.43 ± 1.45 | 1.14 ± 1.03 | 1.38 ± 1.20 |
| *p* | <0.001* | 0.001* | <0.001* | <0.001* |
| Torque (°) | 2.10 ± 1.88 | 3.39 ± 2.69 | 3.15 ± 2.35 | 2.74 ± 2.31 |
| *p* | 0.722 | 0.999 | 0.999 | 0.999 |

Note:
  * Bracket errors were significantly less than 0.5 mm and 2° as indicated by one-sided t-tests.

errors were significantly lower in the PE group for premolars ($p = 0.049$) and the total sample ($p = 0.043$). Torque errors were significantly lower in the FE group for molars ($p = 0.006$).

## DISCUSSION

This study investigated the transfer accuracy of fully enclosed and partially enclosed 3D-printed indirect bonding trays for lingual orthodontics. Our findings partially reject the null hypothesis, as significant differences were observed in certain transfer accuracies between the two tray designs.

The use of a DLP 3D printer in this study was driven by its superior accuracy compared to other technologies, such as liquid crystal display and stereolithography (*Sim et al., 2024*). DLP printers project a complete image of each layer simultaneously, ensuring high resolution and consistent precision across the tray. Furthermore, the choice of rigid resin as the printing material stems from its reduced susceptibility to vertical bracket positioning errors due to uneven finger pressure during transfer. Our preliminary laboratory experiments with flexible resins revealed their inability to securely hold lingual brackets

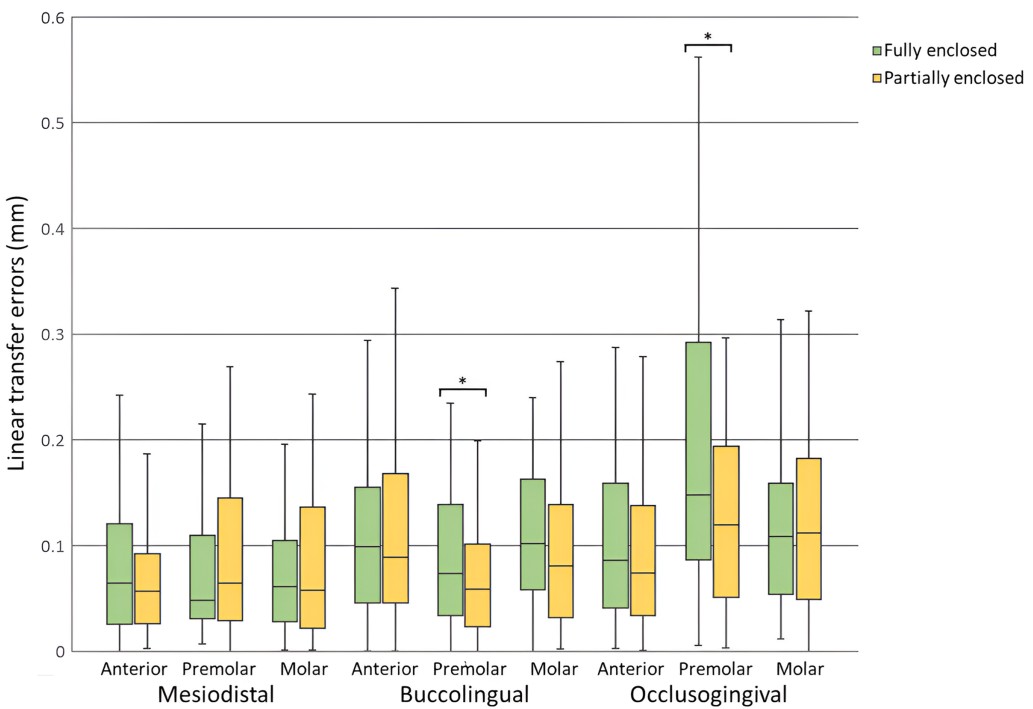

**Figure 3** **Linear transfer errors for fully enclosed and partially enclosed tray designs across all tooth types.** *Statistically significant difference between tray designs ($p < 0.05$).

within the tray structure. *Palone et al. (2023)* observed similar issues in their research and resorted to stabilizing brackets with wax and silicone in flexible tray designs, underscoring the limitations of flexible materials for this application. By contrast, the rigidity of the selected resin in this study obviates the need for such additional steps, improving overall efficiency and accuracy.

In this study, fully enclosed trays were printed at a 110° build angle to optimize gingival wall support, while partially enclosed trays were printed at a 180° build angle to facilitate printing efficiency. Although the build angle may influence the accuracy of indirect bonding trays, previous studies have primarily evaluated the effect of printing orientation on transfer models, not on directly printed trays (*Süpple et al., 2021*; *Zhu et al., 2025*). Therefore, a 180° orientation was selected in this study, in accordance with the approach adopted by *Fiorillo et al. (2023)* and *Karabiber & Eglenen (2024)*, who printed trays in a similar horizontal position. In addition, the i700 intraoral scanner used in this study was selected based on its high scanning accuracy and reproducibility, which are critical for generating reliable virtual models for bracket placement (*Falih & Majeed, 2022*). The Autolign software provided an integrated platform for precise virtual lingual bracket positioning guided by a virtual archwire, minimizing subjective error during setup. For transfer error measurement, Meshmixer software (Autodesk, San Francisco, CA, USA) was utilized because it allows detailed manipulation of three-dimensional models and offers
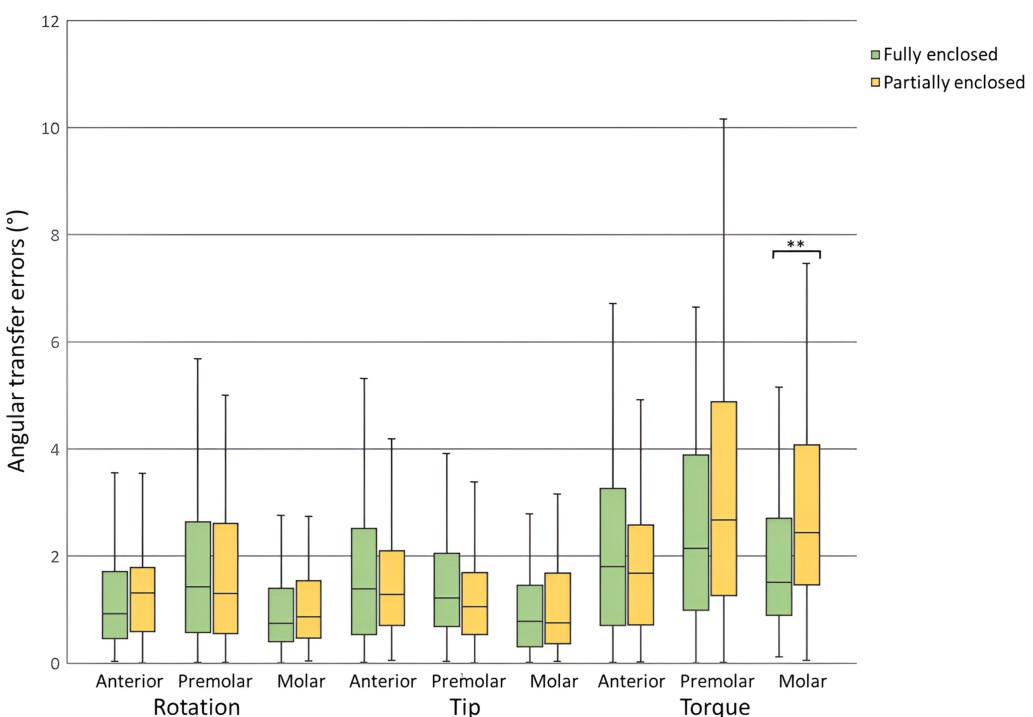

**Figure 4** Angular transfer errors for fully enclosed and partially enclosed tray designs across all tooth types. **Statistically significant difference between tray designs ($p < 0.01$).

**Table 5** Results of two-sided t-tests comparing bracket transfer errors ($p$-values) between fully and partially enclosed tray designs.

|  | Anterior | Premolar | Molar | Total |
|---|---|---|---|---|
| Mesiodistal (mm) | 0.262 | 0.292 | 0.749 | 0.853 |
| Buccolingual (mm) | 0.441 | 0.042* | 0.321 | 0.047* |
| Occlusogingival (mm) | 0.154 | 0.049* | 0.847 | 0.043* |
| Rotation (°) | 0.810 | 0.491 | 0.576 | 0.940 |
| Angulation (°) | 0.361 | 0.337 | 0.941 | 0.288 |
| Torque (°) | 0.274 | 0.555 | 0.006* | 0.235 |

**Note:**
* Statistically significant difference between tray designs ($p < 0.05$).

customizable coordinate axes essential for accurate assessment of both linear and angular discrepancies.

The significantly lower bonding time of PE trays compared to FE trays is attributed to the simplified design of the PE trays, which expose the gingival bracket wings, allowing for easier tray removal. However, the bonding times for both tray types were still longer than those reported by *Anh et al. (2024)* and *Czolgosz, Cattaneo & Cornelis (2021)*, who documented bonding times of approximately 21 and 13 min for lingual and labial brackets,

respectively. The extended bonding time in our study can be partly explained by the additional step required to trim the rigid tray material surrounding the brackets after bonding to facilitate tray removal. Additionally, there is a possibility of tray material co-polymerizing with the adhesive during the light-curing process, necessitating more meticulous cleaning to remove any residual material. This highlights a trade-off between the stability offered by rigid materials and the clinical efficiency achievable with more flexible alternatives.

The improved buccolingual and occlusogingival transfer accuracy observed in the PE trays may stem from the reduced likelihood of the tray material contacting uneven tooth surfaces or undercuts, allowing for complete seating of the tray. In contrast, the fully enclosed design, by encapsulating the brackets entirely, can encounter interference when the brackets are not perfectly seated within the tray. This interference may cause the tray to be lifted or improperly seated on the tooth surface, potentially compromising the transfer process.

The higher torque transfer accuracy of the FE design can be attributed to its superior bracket retention. By encapsulating the brackets completely, the FE design reduces the likelihood of buccolingual rotation of the brackets within the tray during the bonding process. This increased stability ensures that the brackets maintain their intended orientation, particularly in the torque dimension, which is sensitive to even minor deviations. In contrast, the partially enclosed design, while advantageous for overall tray seating, may offer less resistance to such rotational forces, potentially leading to inaccuracies in torque transfer.

The reported higher linear transfer accuracy compared to angular accuracy in this study aligns with previous research on labial brackets (*Niu et al., 2021*; *Bachour, Klabunde & Grünheid, 2022*) and lingual brackets (*Schubert et al., 2013*; *Anh et al., 2024*). This consistency may be due to the rounding effect introduced during scanning, possibly obscuring the precise definition of bracket edges, complicating the determination of the bracket axis and thus increasing the margin of error in angular measurements. Among angular dimensions, torque consistently exhibits the least accuracy, as noted both in this study and earlier reports (*Schubert et al., 2013*; *Niu et al., 2021*; *Bachour, Klabunde & Grünheid, 2022*; *Anh et al., 2024*). This could be attributed to the susceptibility of brackets to buccolingual rotation, which is more likely to occur when the brackets interfere with tooth surfaces during seating due to undercuts or polymerization shrinkage of the tray material.

When comparing the findings of this study with previous research on 3D-printed flexible trays for labial brackets, certain similarities and differences emerge. The *in vivo* study by *Niu et al. (2021)* found mesiodistal and rotation errors of 0.07 mm and 1.22°, respectively, which were comparable to the current study, but reported higher buccolingual, occlusogingival, angulation, and torque errors at 0.13 mm, 0.19 mm, 2.25°, and 3.14°, respectively. Similarly, *Bachour, Klabunde & Grünheid (2022)* observed mesiodistal and buccolingual errors of 0.10 mm, which were comparable, but noted greater inaccuracies in other dimensions, with occlusogingival, rotation, angulation, and torque errors at 0.18 mm, 2.47°, 2.00°, and 2.55°, respectively. The lower transfer accuracy in

these earlier studies may be attributed to the flexibility of the indirect bonding trays used. On the other hand, *Schwärzler et al. (2023)* evaluated 3D-printed rigid trays and reported lower mesiodistal and angular errors at 0.03 mm and 0.26°, respectively, similar occlusogingival errors at 0.12 mm, and higher buccolingual errors at 0.16 mm.

However, the inferior accuracy measured in this *in vivo* study compared to previous *in vitro* studies on labial brackets, as reported by *Koch et al. (2022)* and *Hoffmann et al. (2022)*, can likely be attributed to the controlled experimental conditions, which avoided confounding factors such as restricted intraoral access and the presence of saliva. *Koch et al. (2022)* demonstrated consistently high transfer accuracy across all dimensions, with buccolingual, occlusogingival, rotation, angulation, and torque errors measured at 0.02 mm, 0.06 mm, 0.63°, 0.47°, and 0.52°, respectively. Similarly, *Hoffmann et al. (2022)* reported lower angular errors, including 0.41° for rotation, 0.50° for angulation, and 0.66° for torque. In contrast, the *in vitro* study by *Eglenen & Karabiber (2024)* on labial brackets reported consistently higher linear errors compared to the present study, with mesiodistal, buccolingual, and occlusogingival errors of 0.16, 0.17, and 0.28 mm, respectively. However, their findings showed comparable angular errors, with rotation and torque errors of 1.76° and 2.59°.

The results of this study align with and build upon previous research on the transfer accuracy of lingual brackets. *Schubert et al. (2013)* and *Anh et al. (2024)* reported comparable linear transfer accuracies, ranging from 0.10 to 0.13 and 0.06 to 0.12 mm, respectively. *Anh et al. (2024)* observed similar angular errors, ranging from 1.28° to 2.96°. *Schubert et al. (2013)*, however, reported higher rotation and angulation errors of 2.29° and 2.31°, possibly due to the transfer of brackets onto single teeth, which lacked anatomical landmarks for precise alignment. Conversely, they reported slightly lower torque errors of 2.20°, likely due to better bracket fit within the prefabricated bracket housing.

The findings of this study provide valuable insights for clinicians in selecting 3D-printed indirect bonding trays for lingual orthodontics. The FE tray design, while providing better torque transfer accuracy, is more complex to fabricate, requires longer bonding times, and does not demonstrate a clear overall advantage in transfer precision. In fact, it performs less effectively in buccolingual and occlusogingival accuracy compared to the PE design. Given these considerations, the PE design emerges as the more practical choice for most clinical cases due to its simplicity, efficiency, and satisfactory accuracy across multiple dimensions. To address the torque errors observed with the PE design, clinicians can consider supplementing the tray with additional stabilizing materials, such as wax or silicone, to secure brackets during bonding, particularly for molar brackets. This approach may enhance torque transfer accuracy while retaining the advantages of the PE design in terms of ease of use and reduced chair time.

This study has several limitations that warrant consideration. First, the study did not evaluate the performance of flexible tray materials, which could offer advantages such as better adaptability to tooth surfaces, improved tolerance to undercuts, and easier tray removal, particularly in cases with significant crowding. Second, the study did not assess the impact of design parameters such as offset, tray thickness, or build angle during the 3D printing process on tray accuracy. Although this study did not specifically assess the effect

of incisor crowding, previous research by *Jungbauer et al. (2021)* demonstrated that severe crowding can adversely affect bracket transfer accuracy, especially when using rigid tray materials. However, any teeth with excessive crowding that caused bracket loosening within the tray were excluded from bonding, and the corresponding teeth were omitted from tray segmentation to avoid interference with tray insertion and removal.

Additionally, the study utilized lingual brackets, resin materials, an intraoral scanner, and a 3D printer from specific manufacturers, which may limit the generalizability of the findings to other brands or equipment with varying specifications and performance characteristics. While the comparison aimed to isolate the effect of tray design, baseline characteristics such as age, gender, treatment type, and crowding were comparable between groups, supporting internal validity. However, the absence of randomization introduces a risk of selection bias that should be considered when interpreting the findings. Furthermore, the sample size calculation was based on the largest angular deviation (torque error) from previous studies to ensure adequate power for detecting clinically significant discrepancies in the most challenging dimension and overall assessment. However, the number of brackets in certain subgroups, particularly premolars, may have been insufficient to reliably detect smaller differences, which should be considered when interpreting the subgroup analyses. Future research addressing these gaps would further refine the understanding of indirect bonding tray designs and their clinical applications.

## CONCLUSIONS

In summary, both fully and partially enclosed 3D-printed indirect bonding trays achieved clinically acceptable accuracy in transferring linear bracket positions. Angular bracket positions, specifically rotation and angulation, were also transferred with clinically acceptable accuracy. However, torque transfer remains a challenge for both tray designs. The partially enclosed tray design demonstrated significantly lower buccolingual and occlusogingival errors for premolars and the total sample, while the fully enclosed tray design achieved significantly lower torque errors for molars. Despite this, the fully enclosed design was more complex to fabricate, required longer bonding times, and provided no significant overall advantage in accuracy, making the partially enclosed design the preferred choice. To address torque errors in molar brackets with the partially enclosed design, the use of supplementary stabilization materials is recommended.

### Funding
The author received no funding for this work.

### Competing Interests
Viet Anh Nguyen is the owner of Viet Anh Orthodontic Clinic, Hanoi.

## Author Contributions

- Viet Anh Nguyen conceived and designed the experiments, performed the experiments, analyzed the data, prepared figures and/or tables, authored or reviewed drafts of the article, and approved the final draft.

## Human Ethics

The following information was supplied relating to ethical approvals (*i.e.*, approving body and any reference numbers):

This study was conducted in compliance with ethical standards and approved by the Hanoi Medical University Institutional Ethical Board Review (approval no. HMUIRB970).

## Data Availability

All data generated or analysed during this study are available in the Supplemental File.

## Supplemental Information

Supplemental information for this article can be found online at http://dx.doi.org/10.7717/peerj.19612#supplemental-information.

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
