# Peer review of "Comparative analysis of lingual bracket transfer accuracy using fully versus partially enclosed 3D-printed indirect bonding trays: an in vivo study"

_PeerJ, doi:10.7717/peerj.19612_

## Round 0.1 · original submission · Major Revisions

Please revise your manuscript in accordance with the reviewers' comments, giving particular attention to the concerns raised by Reviewers 2 and 3.

·

Basic reporting

Thank you for this valuable work and for the understandable language.

Experimental design

Please provide a more detailed description of the inclusion and exclusion criteria in the Materials and Methods section. Additionally, please ensure that the method includes the brand of brackets used in the study.

Validity of the findings

no comment

Additional comments

In the Discussion section, a more thorough analysis regarding the Materials and Methods would be beneficial. It is also important to discuss the brand of the intraoral scanner employed, as well as the 3D printer and software program used in the study, in the context of existing literature.

Furthermore, to enhance understanding of the transfer trays used, it would be helpful to include an additional figure from a different angle.

Reviewer 2 ·

Basic reporting

• Exact p-values should be reported either in the tables or in the main text.

Experimental design

• It remains unclear how patients were recruited. How many were screened for eligibility, how many had to be excluded and what were the reasons for exclusion. I recommend adding a detailed flow chart of the recruitment (CONSORT flow diagram).
• There is no information on how randomization was performed.
• The author should report where the trial was registered in advance.
• How were the teeth cleaned before etching?
• I recommend reporting the Shore hardness of the rigid print material.
• How long were the trays cleaned and cured after printing?
• How was excess resin removed after bracket bonding?
• The matching process needs to be described in more detail.

Validity of the findings

• “The significance level was set at α = 0.05”?
• The information on the evaluation of the ICC is not sufficient. Who was the second person to assess the data? In addition, it is strongly recommended for this type of study to assess the intraclass reliability by having some of the measurements re-rated by the same principal investigator. It is also recommended that the entire matching process be repeated to assess the reliability of the measurements (Jungbauer et al. 2021, https://doi.org/10.3390/app11136013 ).
• ICC results are not reported.
• The model, type and definition of the ICC used should be reported.
• According to material and method bonding time has not been evaluated but is discussed. Either the assessment needs to be reported (definition of bonding time, how it was assessed) or the section in the discussion must be removed.
• Jungbauer et al. 2021 (https://doi.org/10.3390/app11136013 ) found that severe crowding of the incisors is likely to have an impact on transfer accuracy, especially when hard trays are used. Therefore, incisor crowding should be assessed to ensure that patients with severe crowding (Little’s Irregularity Index >7 ) were not included in this study.

Additional comments

• The design of the bonding tray is not entirely clear to me. Were they divided into segments?

Reviewer 3 ·

Basic reporting

Introduction:
- Incomplete literature/insufficient field background (in particular regarding bonding tray designs)

Results:
- the reported ranges are somewhat incomprehensible and not congruent with the illustrations provided

Discussion
- make sure to point out which studies investigated lingual or vestibular bonding
- it should be discussed why “additional trimming” was performed and how it could affect the results

Figures and tables:
- the axis labeling is missing from the illustrations

- table 3: units are missing; it is currently not clear from the table what the asterisk signifies (which comparison)

Experimental design

- the sample size used is too small for some comparisons, such as between different tooth groups or specifically premolars. Calculating the sample size based on the largest observed deviation of another study may lead to insufficient statistical power, making it difficult to detect meaningful differences reliably. The rationale and limitations behind this approach should be clearly explained

- you are refering to a workflow from another paper, make sure to name at least the most important details, such as software used for virtual bracket bonding

- how was low and moderate crowding defined?

- Intraoral lingual scanning of brackets is challenging due to reflections and limited accessibility. Previous studies have demonstrated that such distortions typically cause measurement errors that are similar in magnitude to the dimensions being evaluated. To reliably assess the accuracy and validity of intraoral lingual scans of brackets and their superimposition, the reporting of data supporting the used methodology is required. Please provide the data (e.g. Bland-Altman analyses and analysis of variance) evaluating systematic and random measurement errors.

- studys have shown that changing the build angle has an influence on the accuracy of indirect bonding; the rationale “to reduce printing time” is not suitable for a scientific study, where such aspects present a serious methodological issue

- in line 126 - did you scan dental models or patients?

- how were the bracket designs derived? Please report the full parameters for both trays

Validity of the findings

no comment (see 2.)

---

## Round 0.2 · Major Revisions

The manuscript has been considerably improved after the revisions; however, important issues raised by Reviewer 2 still need to be addressed before it can be considered further.

·

Basic reporting

After the revision, the article is much more understandable and detailed.

Experimental design

Information gaps in the material method have been eliminated.

Validity of the findings

This article will be useful on this subject.

Additional comments

Thank you for the revision.

Reviewer 2 ·

Basic reporting

Firstly, I would like to thank the author for considering and discussing my comments. However, I still have some questions:
1. The results of the time needed for bonding measurements should be reported in the Results section.

Experimental design

2. How was randomization performed? According to the point-by-point response, no randomization was performed, yet the following sentence appears in the Materials and Methods section of the manuscript: 'Patients were randomized into two groups'. In other words, if no randomization was performed, how were the patients allocated to one of the two groups? Without proper randomization, there would be significant unnecessary bias, limiting the scientific value of this investigation.
3. Were only lower jaws included in this study, or both?

Validity of the findings

-

---

## Round 0.3 · accepted · Accept

I am satisfied that all the reviewers' concerns have been adequately addressed, and the manuscript is suitable for publication in its current form.